# Synthesising a minimal cell with artificial metabolic pathways

Minoru Kurisu [1], Ryosuke Katayama[1], Yuka Sakuma[1], Toshihiro Kawakatsu[1], Peter Walde [2] & Masayuki Imai [1✉]

A "synthetic minimal cell" is considered here as a cell-like artificial vesicle reproduction system in which a chemical and physico-chemical transformation network is regulated by information polymers. Here we synthesise such a minimal cell consisting of three units: energy production, information polymer synthesis, and vesicle reproduction. Supplied ingredients are converted to energy currencies which trigger the synthesis of an information polymer, where the vesicle membrane plays the role of a template. The information polymer promotes membrane growth. By tuning the membrane composition and permeability to osmolytes, the growing vesicles show recursive reproduction over several generations. Our "synthetic minimal cell" greatly simplifies the scheme of contemporary living cells while keeping their essence. The chemical pathways and the vesicle reproduction pathways are well described by kinetic equations and by applying the membrane elasticity model, respectively. This study provides new insights to better understand the differences and similarities between non-living forms of matter and life.

[1] Department of Physics, Graduate School of Science, Tohoku University, 6-3 Aramaki, Aoba, Sendai 980-8578, Japan. [2] Department of Materials, ETH Zürich, Vladmir-Prelog-Weg 5, CH-8093 Zürich, Switzerland. ✉email: imai@bio.phys.tohoku.ac.jp

Living systems are cell-based and reproduce themselves through a network of chemical transformations (metabolism), whereby the offsprings have the same proliferation abilities as the parent structure (recursive reproduction). The chemical network is maintained by numerous proteins that are generated based on sequence information encoded in DNA. The essential metabolic pathways are shown in Fig. 1a; (i) The energy production unit (orange), which synthesises energy currencies from ingredients, (ii) the processing of genetic information unit (green), where DNA is replicated, and proteins are synthesised based on DNA, and (iii) the membrane synthesis unit (blue), where amphiphilic lipid molecules are synthesised using energy currencies and proteins[1,2]. The synthesised lipid molecules are incorporated into the cell membrane, which results in membrane growth and division, i.e., proliferation. Fig 1a is also a representation of Gánti's chemoton model of living systems[3–5]. A living system requires recursive compartment proliferation by regulating compartment membrane area growth, compartment volume growth, as well as compartment deformation, and division processes.

A traditional approach for understanding living systems is to reconstruct the metabolic pathways that synthesise lipids using proteins expressed by DNA inside a vesicular compartment so that vesicle reproduction occurs[6–11]. However, since this approach utilises the heart of a complex living system (i.e., the central dogma of molecular biology), shedding light on the road from non-living forms of matter to living systems is still very challenging. An alternative soft matter approach is to synthesise simplest cell-like systems, also called minimal cells[12,13], whereby vesicles reproduce themselves based on instructions encoded in information molecules using non-biological substances. The first step toward this goal is to construct the chemical pathways which lead to vesicle reproduction[14–21]. An essential key to develop these vesicle reproduction systems into minimal cells is the coupling between synthesis of information molecules and vesicle reproduction[22]. Recently, we developed such a vesicle reproduction system in which vesicle growth and division are coupled with the synthesis of an information polymer[23]. The information polymer is the emeraldine salt form of polyaniline (PANI-ES), which has a characteristic sequence of segments. The sequence is determined by a specific paring between the segment and the membrane molecule when PANI-ES is synthesised on giant unilamellar vesicles (GUVs) composed of sodium bis-(2-ethylhexyl) sulfosuccinate (AOT), using the template polymerisation mechanism[24,25]. In contrast to the biological information polymers (DNA, RNA) that express proteins for synthesising membrane molecules, the information polymer PANI-ES encourages the growth of the GUVs by incorporating membrane molecules (AOT) from the environment into the GUVs. In addition, the GUVs show division, i.e., reproduction, by introducing cholesterol (Chol) to the AOT vesicle membrane due to the coupling between Gaussian curvature and local lipid composition of the membrane[23,26–29]. However, our previous system has no chemical reaction network starting from an energy production unit, and the vesicle reproduction is not sustainable.

In this study, based on the AOT + Chol/PANI-ES vesicle reproduction system developed previously[23], we present an advanced synthetic minimal cell system with an artificial metabolic pathway, which shows sustainable recursive production of vesicles instructed by the information polymer PANI-ES. First, we design artificial metabolic pathways composed of (i) energy production, (ii) synthesis of an information polymer, and (iii) vesicle membrane growth units. By implementing these pathways in the vesicle reproduction system, the energy currencies produced in the energy production unit trigger the template polymerisation of aniline on the AOT vesicle, which results in vesicle membrane growth. Based on the chemical schemes, we construct a kinetic model of the artificial metabolic pathways, which

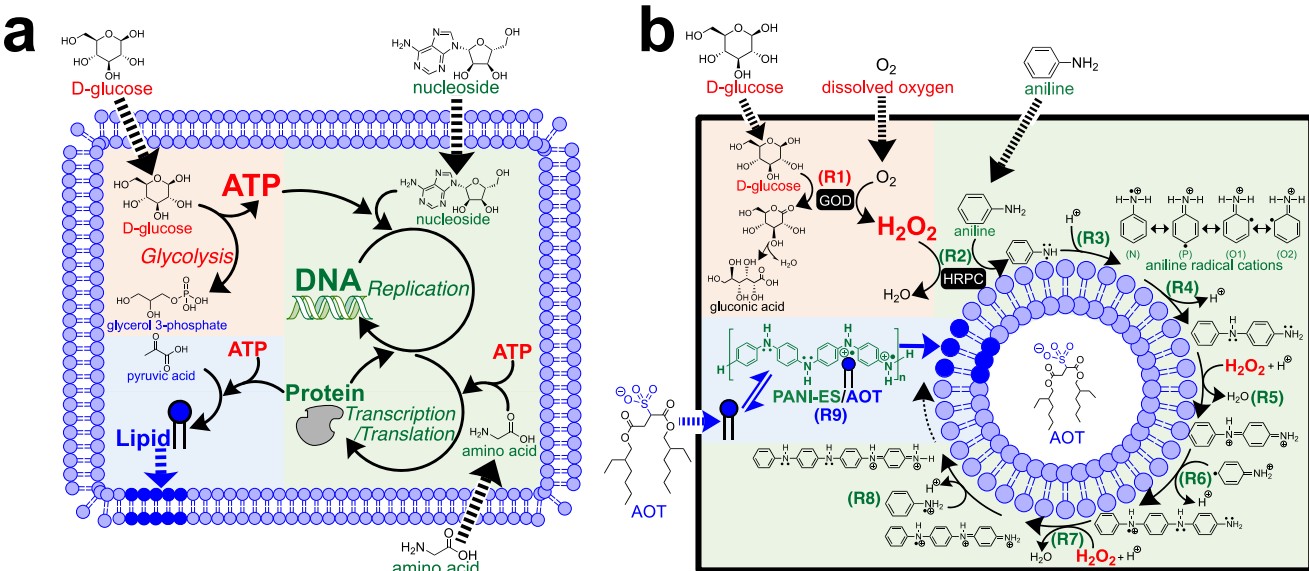

**Fig. 1 Simplified chemical schemes of the essential metabolic transformations in a contemporary living system and in the synthetic minimal cell system presented in this work. a** Deduced essential metabolic pathways of a contemporary living system consisting of three elementary units; energy production (orange), processing of genetic information (green), and synthesis of membrane molecules (blue). Black dashed arrows represent the inflow of ingredients through the membrane, and the blue dashed arrow indicates the incorporation of synthesised lipids into the membrane. Such simplified chemical schemes have been also proposed by others[2–4]. **b** Chemical scheme of the artificial metabolic pathways of our synthetic minimal cell consisting of three elementary units; energy production (orange), synthesis of information polymer (green), and membrane growth (blue). The black dashed arrows towards the inside of the thick rectangle represent the reactants (ingredients) supplied close to the vesicles. Membrane molecules (AOT) are also supplied close to the vesicles (blue dashed arrow), being incorporated into the membrane through PANI-ES (blue paired harpoons). For the detailed reaction steps, (R1)–(R9), and the reaction conditions, see Supplementary Note 1 and **Materials & Methods**, respectively.

quantitatively describes the observed synthesis of the information polymer and vesicle membrane growth. Furthermore, by regulating the supply of ingredients and the osmotic pressure, our synthetic minimal cells realise a recursive reproduction cycle consisting of the following steps: vesicle membrane growth → vesicle deformation → vesicle division → vesicle inflation. Finally, we will discuss the physical background of the synthetic minimal cell coupled to the artificial metabolic pathways.

## Results

**Chemical scheme of the artificial metabolic pathways.** Fig 1b summarises the chemical transformations of the artificial metabolic pathways of our synthetic minimal cell. The transformations are: energy production (R1) (orange), synthesis of the information polymer (R2)–(R8) (green), and membrane growth (R9) (blue) units. This scheme is based on a membrane-assisted enzymatic cascade reaction to obtain the information polymer PANI-ES, as recently described[30]. In the energy production unit (Supplementary Note 1-1), ingredients, D-glucose and $O_2$, taken from the environment, are converted in a redox reaction catalysed by glucose oxidase (GOD) to energy currencies (by-product $H_2O_2$) and gluconic acid as the main product (R1). In the synthesis of the information polymer unit (Supplementary Note 1-2), the obtained $H_2O_2$ oxidises horseradish peroxidase isoenzyme C (HRPC), which then oxidises aniline, forming anilino radicals (R2). The protonation of anilino radicals produces aniline radical cations (shown are the four possible resonance structures) (R3). In the aniline radical cation, the unpaired electron is localised either on the nitrogen atom (N), on the carbon atom in *ortho*-position (two positions, O1 and O2), or in *para*-position (P). Two aniline radical cations react to yield one of several possible dimers (R4)[25,31]. The specificity feature of polyaniline as simple information polymer for our minimal cell system is that on the surface of the vesicle membrane, which is formed from AOT (an amphiphile with a sulfonate head group), the dimerisation and further elongation processes preferentially occur between the unpaired electrons localised at the C-atom in the *para*-position (P) and at the nitrogen atom (N) (*i.e.*, a linear *para*-NC coupled sequence) due to specific hydrogen bonding between the aniline radical cation and AOT (template effect)[25,32–34] (see **Discussion** section). Continuing these regulated elongations to form the aniline tetramer dication yields one repeating unit of PANI-ES with two in oxidised and two in reduced states, (R5)–(R8). In contrast, amphiphiles without sulfonate (or sulfate) head groups lack specific interactions on their membranes and, therefore, limit the proper formation of the monomer sequence during the polymerisation process: the four aniline radical cations connect randomly and form ill-defined polyaniline products, *i.e.*, extensively branched and a mixture of compounds[23,34,35]. Thus, the linear *para*-NC coupling of aniline on the AOT membrane is the origin of the sequence information. In the membrane growth unit (Supplementary Note 1-3), PANI-ES on the AOT vesicles interacts with AOT molecules present in the external solution through specific hydrogen bonding[25,34], resulting in the incorporation of AOT molecules into the vesicle membrane and the subsequent vesicle growth (R9). The integration of these three units is the chemical basis of the artificial metabolic pathway of our synthetic minimal cell.

**Implementation of the artificial metabolic pathways.** To realise the artificial metabolic pathways mentioned above, we first coupled the energy production unit (R1) with the synthesis of the information polymer unit (R2)–(R8). The cascade reaction conditions were first elaborated in a reaction tube following our previous study[30] and then applied to our synthetic minimal cell system (see **Methods**). Inside a reaction tube, the cascade reaction was triggered by adding a solution of the enzyme GOD to a suspension of AOT large unilamellar vesicles (LUVs) (prepared in 20 mM $NaH_2PO_4$ solution, pH = 4.3) containing aniline, HRPC, and D-glucose. The energy currency, $H_2O_2$, is produced as a by-product of the GOD-catalysed oxidation of D-glucose with dissolved $O_2$. $H_2O_2$ is subsequently used for the HRPC-catalysed polymerisation of aniline on the AOT vesicles. The formation of PANI-ES in the presence of AOT LUVs was confirmed by analysing the appearance of a characteristic maximal absorption at $\lambda \sim 1000$ nm in the UV/Vis/NIR absorption spectrum due to delocalised polarons (Fig. 2a, red solid line), which agrees well with that of the HRPC-catalysed formation of PANI-ES obtained with the direct addition of $H_2O_2$ (Supplementary Note 2-1). For reference, we show a typical absorption spectrum for a mixture of random sequence polyaniline synthesised in the presence of DOPC (1,2-dioloeyl-*sn*-glycero-3-phosphocholine) LUVs (blue dashed line) and in the absence of vesicles (black dotted line), where no characteristic absorption in the NIR region of the spectrum is observed (Fig. 2a, dashed and dotted lines). The synthesis of polyaniline in the presence of AOT LUVs or DOPC LUVs with the cascade reaction scheme was also examined by Raman spectroscopy (Fig. 2b). The Raman spectrum of the cascade reaction products obtained in the presence of AOT LUVs has a clear $\nu(C\sim N^{\bullet+})_p$ peak at $\sim 1345$ cm$^{-1}$ due to delocalised polarons of PANI-ES, which agrees well with previous reports on the analysis of PANI-ES[36–38]. In contrast, no such characteristic Raman band was observed for the products obtained from the reaction run in the presence of DOPC LUVs, which is fully consistent with the analysis of the UV/Vis/NIR absorption spectrum. Based on the success in the implementation of the energy production unit in the presence of AOT LUVs, we examined the synthesis of PANI-ES from D-glucose on individual AOT GUVs. In the presence of AOT GUVs, aniline was polymerised using the optimised cascade reaction conditions. The localisation of polyaniline on AOT GUVs was confirmed by micro-Raman spectroscopy. The micro-Raman spectrum (Fig. 2c) obtained from an AOT GUV after completion of the reaction (point "A" in Fig. 2d) shows the characteristic PANI-ES Raman peak as shown in Fig. 2b, indicating the production of PANI-ES on the AOT GUV. Here, the low energy resolution of the Raman spectrum is due to the weak scattering intensity from the AOT/PANI-ES GUV. Using the characteristic peak at 1345 cm$^{-1}$, we constructed a two-dimensional Raman map with 0.5 μm resolution on the AOT GUV (see **Methods**), as shown in Fig. 2d. The very good agreement between the optical microscope image and the Raman image indicates that PANI-ES is distributed nearly homogeneously on the AOT GUV surface.

The time evolution of the synthesis of PANI-ES in the presence of AOT LUVs using the cascade reaction was monitored by UV/Vis/NIR spectroscopy following the absorption peak at $\lambda \sim 1000$ nm (Fig. 2e)[25,39], which showed that the PANI-ES synthesis was completed within a time scale of ~150 s due to the consumption of the ingredients in the reaction tube.

In a second step, we demonstrated vesicle membrane growth coupled with the artificial metabolic pathway (R9). We prepared AOT GUVs in an aqueous solution that is optimal for the synthesis of PANI-ES with the cascade reaction, containing 4.0 mM aniline, 0.92 μM HRPC, and 1.0 μM GOD in 20 mM $NaH_2PO_4$ solution (pH = 4.3). Then, a 100 mM D-glucose solution containing 20 mM AOT micelles was micro-injected to a target AOT GUV by using a double micro-injection technique (see **Methods**). The initially spherical AOT GUV began to grow with a deformation into a prolate shape after ~50 s from the start

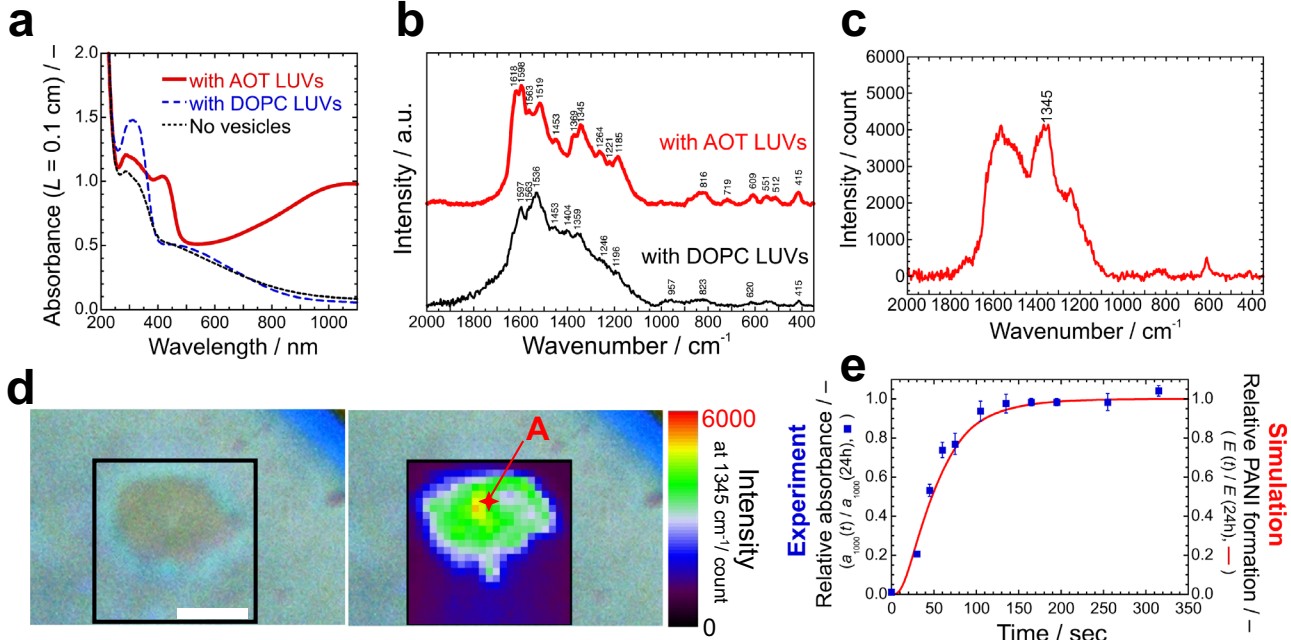

**Fig. 2 Synthesis of PANI-ES on the surface of AOT vesicles. a** UV/Vis/NIR absorption spectra of the polymerisation products. The absorption spectra of the products were obtained by polymerisation of aniline in the presence of AOT LUVs under the optimised cascade reaction conditions (red solid line, see **Methods**). For reference, the absorption spectra of the products obtained from the polymerisation of aniline either in the presence of DOPC LUVs (blue dashed line) or in the absence of any LUVs (black dotted line) are also shown. All spectra were recorded 24 h after the start of the reactions at $T$ ~25 °C. **b** Raman spectra of the products obtained by the polymerisation of aniline in the presence of either AOT LUVs (red thick line) or DOPC LUVs (black solid line) under the optimised cascade reaction conditions. The spectra were recorded at $T$ ~25 °C after ~24 h from the start of the reaction, see **Methods**. **c** Micro-Raman spectra of an AOT/PANI-ES GUV obtained under the optimised cascade reaction conditions. The sampling point is shown as point "A" in **d**. The spectrum was recorded after ~24 h from the start of the reaction at $T$ ~25 °C. The laser spot size was 520 nm in diameter and 680 nm in depth. **d** Spatial Raman mapping of an AOT/PANI-ES GUV using the 1345 cm⁻¹ peak, which is characteristic for the delocalised polarons of PANI-ES. Microscope bright field image (left) and Raman mapping image (right) of an AOT/PANI-ES GUV are shown. The scanning was started after ~24 h from the start of the reaction at $T$ ~25 °C. The shadow in the upper right is caused by the wall of the glass tube. Length of the scale bar: 5 μm. **e** Time-dependence of the relative absorbance at $\lambda = 1000$ nm of the UV/Vis/NIR absorption spectra obtained from the polymerisation products of aniline in the presence of AOT LUVs under the optimised cascade reaction conditions (blue squares). The absorbance at $\lambda = 1000$ nm ($a_{1000}(t)$) is normalised by the absorbance after 24 h from the start of the reaction ($a_{1000}(24\,\text{h})$). The red line is the theoretical prediction obtained by the kinetic model (model-2, see Supplementary Note 3-3(ii)). The error bars indicate standard deviations estimated from three different experiments.

of injection (Fig. 3a and Supplementary Movie 1), which coincided with the time scale for the formation of PANI-ES (Fig. 2e). The prolate vesicle elongated with elapse of time by uptake of AOT molecules from the external solution. We estimated the vesicle surface area by approximating the vesicle shape as axisymmetric prolate shape. The time evolution of the surface area, $A(t)$, for the optimised cascade reaction conditions is plotted as red circles in Fig. 3b, where the vesicle surface area is normalised by the surface area of the initially spherical vesicle, $A(0)$. This experiment was carried out six times, each time using a different vesicle sample. The target vesicles showed exponential growth, although the growth rate was almost half compared with that triggered by the direct addition of $H_2O_2$ in the absence of D-glucose and GOD (Supplementary Note 2-2). For reference, we micro-injected AOT micelles to AOT GUVs under conditions where no polymerisation took place, see Fig. 3b. For all reference conditions, vesicle growth was suppressed. Thus, we have succeeded in achieving vesicle membrane growth by coupling the artificial metabolic pathways (R1) – (R9).

**Kinetic model of the artificial metabolic pathways.** One of the advantages of the synthetic minimal cell study is that it allows a complete description of the reaction pathways. Here we develop a kinetic model of the artificial metabolic pathway based on the chemical scheme (R1) – (R9) in Fig.1b, where we used a reduced

model reaction scheme, Eqs. (1)–(5) (see Supplementary Note 3-1).

$$R_a + R_b + X \xrightarrow{v_1} Z + X \tag{1}$$

$$2S + Y + Z \xrightarrow{v_2} 2S^* + Y \tag{2}$$

$$2S^* + 2A_v^o \xrightarrow{v_3} P_2 + 2A_v^o \tag{3}$$

$$S^* + P_n + Z + 2A_v^o \xrightarrow{v_4} P_{n+1} + 2A_v^o \tag{4}$$

$$A_m + P_n \xrightarrow{v_5} A_v^o + P_n \tag{5}$$

The energy production reaction Eq. (1) expresses the GOD ($X$)-catalysed oxidation of D-glucose ($R_a$) using dissolved $O_2$ ($R_b$), which produces the energy currency $H_2O_2$ ($Z$). In the production of activated monomer reaction Eq. (2), the HRPC ($Y$)/$H_2O_2$ ($Z$)-catalysed oxidation of aniline ($S$) produces the anilino radical (not shown), which is followed by protonation to obtain the aniline radical cation ($S^*$), *i.e.*, the activated monomer. The initiation reaction of the information polymer synthesis Eq. (3) describes the reaction between two aniline radical cations ($S^*$) to form an aniline dimer ($P_2$) on the surface of the outer leaflet of AOT vesicle ($A_v^o$). In the propagation reaction of the information polymer synthesis Eq. (4), the PANI-ES chain ($P_n$) is oxidised

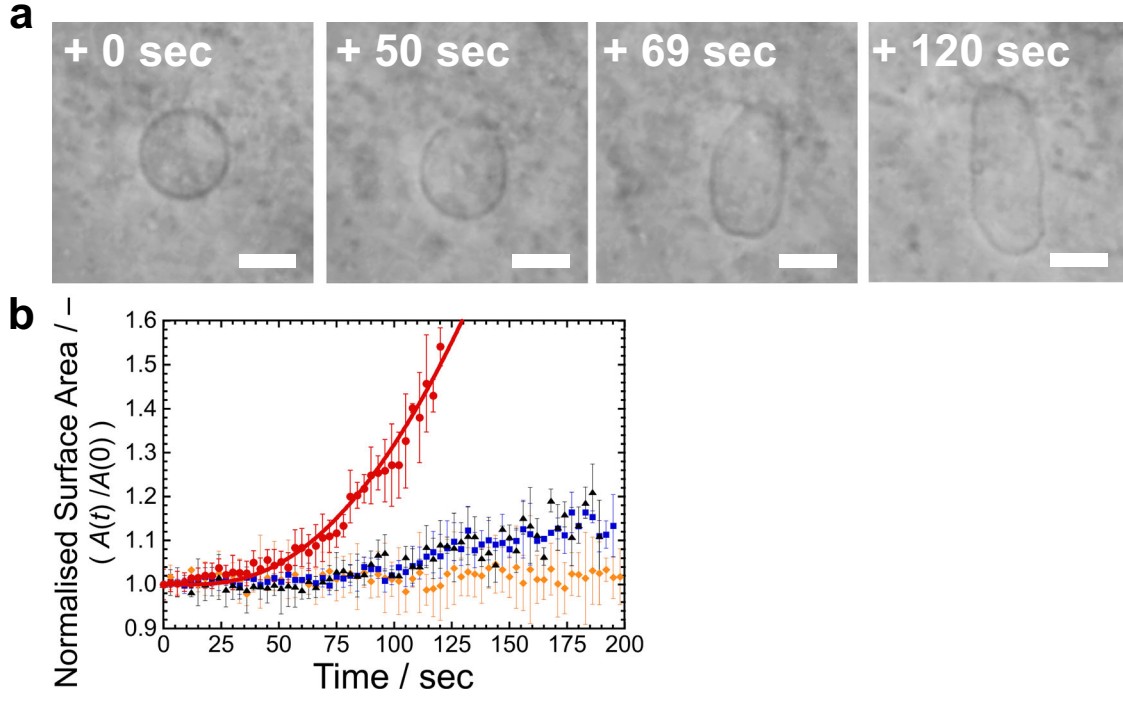

**Fig. 3 Membrane growth of AOT GUVs promoted by PANI-ES on the vesicle surface. a** Phase contrast light microscopy images of an AOT GUV during polymerisation of aniline under the optimised cascade reaction conditions combined with the micro-injection of 20 mM AOT micellar solution containing 100 mM D-glucose (see **Methods**). The elapsed time after starting the micro-injection is indicated in each image. Length of the scale bar: 10 μm. See also **Supplementary Movie 1**. **b** Membrane growth of AOT GUVs in response to the micro-injection of AOT micelles under various polymerisation conditions. Red circles: Membrane growth of AOT GUVs under the optimised cascade reaction conditions. Red solid line: Theoretical prediction based on the kinetic model (model-1, see Supplementary Note 3-3(i)). Blue squares: D-glucose was not contained in the micro-injection solution (otherwise, the optimised cascade reaction conditions were applied), i.e., lack of ingredients for $H_2O_2$ production. Orange diamonds: Aniline, HRPC, and GOD were not contained in the AOT GUV suspension (otherwise, the optimised cascade reaction conditions were applied), i.e., lack of enzymes for the production of $H_2O_2$ and lack of monomer and enzyme for the polymerisation reaction to occur. Black triangles: D-glucose was not contained in the micro-injection solution, and aniline, HRPC, and GOD were not contained in the AOT GUV suspension (otherwise, the optimised cascade reaction conditions were applied), i.e., only an AOT micellar solution was supplied to AOT GUVs in 20 mM $NaH_2PO_4$ solution (pH = 4.3). The surface area of each GUV at time $t$ ($A(t)$) was normalised by the initial value at $t = 0$ ($A(0)$). The error bars indicate standard deviations estimated from four to six different experiments.

with $Z$ to produce an oxidised PANI-ES chain ($P_n^*$, not shown), and then an $S^*$ reacts with $P_n^*$ to produce an elongated PANI-ES chain $P_{n+1}$ on $A_v^o$. The growth of the vesicle membrane coupled with the information polymer, Eq. (5), describes the transformation of free membrane molecule ($A_m$) present in the external solution into $A_v^o$ through $P_n$. Assuming steady-state conditions and a homogeneous distribution of the reactants on the vesicle surface, we obtained simultaneous differential equations for the reduced model reactions (see Supplementary Note 3-1, 3-2, and 3-3(i)),

$$\frac{d[Z(t)]}{dt} = v_1(t) - \left( v_2(t) + v_4(t) \right)[A(t)] \tag{6}$$

$$\frac{d[S^*(t)]}{dt} = \left( 2\,v_2(t) - 2\,v_3(t) - v_4(t) \right)[A(t)] \tag{7}$$

$$\frac{d[P(t)]}{dt} = v_3(t)[A(t)] \tag{8}$$

$$\frac{d[E(t)]}{dt} = \left( 2\,v_3(t) + v_4(t) \right)[A(t)] \tag{9}$$

$$\frac{d\left[A_v(t)\right]}{dt} = v_5(t)[A(t)], \tag{10}$$

where

$$v_1(t) = \frac{[X]_0}{\frac{1}{k_{1,a}} + \frac{1}{k_{1,b}[R_a(t)]} + \frac{1}{k_{1,c}[R_b]}} \tag{11}$$

$$v_2(t) = \frac{[Y]_0}{\frac{1}{k_{2,a}[Z(t)]} + \frac{k_{2,b}+k_{2,c}}{k_{2,b}k_{2,c}[S(t)]}} \tag{12}$$

$$v_3(t) = k_3 n_v \left( \frac{K_{S^*}[S^*(t)]}{1 + K_{S^*}[S^*(t)] + K_Z[Z(t)]} \right)^2 \tag{13}$$

$$v_4(t) = k_4 n_v \frac{[P(t)]}{[A(t)]} \frac{K_Z[Z(t)] \cdot K_{S^*}[S^*(t)]}{\left( 1 + K_Z[Z(t)] + K_{S^*}[S^*(t)] \right) \left( K_Z[Z(t)] + K_{S^*}[S^*(t)] \right)} \tag{14}$$

$$v_5(t) = k_5 \left[ A_m \right] \frac{[E(t)]}{[A(t)]} \tag{15}$$

$$[A(t)] = \frac{1}{2} \left[ A_v(t) \right] N_A\, a_v. \tag{16}$$

Eq. (6) describes the time evolution of the energy currency $Z$, consisting of the rate equations for the production of $Z$, $v_1(t)$, and for the consumption of $Z$, $v_2(t)$ and $v_4(t)$, in the processes of

activating the monomer and of propagating the information polymer, respectively. $[A(t)]$ is the time dependence of the vesicle surface area exposed to the external solution per unit volume given by Eq. (16) ($[A_v]$: amount of membrane forming molecule (AOT) per unit volume, $N_A$: Avogadro constant, and $a_v$: surface area per single membrane molecule). The rate equations for the production of $Z$, $v_1(t)$ given by Eq. (11), is expressed by the two-substrate ping-pong mechanism[40,41]. The rate equations for the consumption of $Z$, $v_2(t)$ given by Eq. (12), is expressed by the irreversible ping-pong mechanism[42], and $v_4(t)$ given by Eq. (14) is expressed by the irreversible ping-pong mechanism combined with the Langmuir-Hinshelwood mechanism[43]. Eq. (7) presents the time evolution of the activated monomer $S^*$ expressed by the rate equations for the production of $S^*$, $v_2(t)$, and for the consumption of $S^*$, $v_3(t)$ and $v_4(t)$, in the processes of initiation and propagation of the polymerisation, respectively. The rate equation for the formation of $P_2$, $v_3(t)$ given by Eq. (13), is expressed by the Langmuir-Hinshelwood mechanism[43]. Eqs. (8) and (9) describe (i) the time evolution of the total number of polymer chains ($P = \sum_{n=1}^{\infty} N_n$, $N_n$: number of vesicle surface-localised polymer (PANI-ES) chains with a degree of polymerisation of $n \geq 2$), and (ii) the time evolution of the monomeric units of the polymer ($E = \sum_{n=1}^{\infty} n N_n$), respectively. Eq. (10) describes the increase of the number of membrane molecules in the vesicle membrane ($[A_v(t)]$) by uptake of membrane molecules from the external solution ($[A_m]$). The binding of membrane molecules present in the external solution to the polymer on the vesicle membrane (formation of AOT–PANI-ES complex) decreases the hydrophilicity of the bound membrane molecules, which results in their incorporation into the membrane (Supplementary Note 1-3). This process is expressed by Eq. (15). In Eqs. (11)–(14), the reaction rate constants, $k_{1,i}$ ($i$ = a, b, and c)[44] and $k_{2,j}$ ($j$ = a, b, and c)[42] are obtained from the literature. $K_{S^*}$ and $K_Z$ are the adsorption equilibrium constants of $S^*$ and $Z$ on the vesicle membrane, respectively, assumed to have the same value as that of the anilinium cation on the AOT membrane, estimated by MD simulations[33]. The reaction rate constants for the polymerisation reaction, $k_3$ and $k_4$ in Eqs. (13) and (14) were determined experimentally from the time evolution of the PANI-ES synthesis on AOT LUVs triggered with the direct addition of $H_2O_2$ (control experiment-0: Supplementary Note 2-1 and 3-3(iii)), and $k_5$ of Eq. (15) was determined experimentally from the growth profile of AOT GUVs coupled with the synthesis of PANI-ES (control experiment-1: Supplementary Note 2-2 and 3-3(iv)). Therefore, the kinetic model used has no free parameters. All parameters are listed in Supplementary Table 1.

We compared predictions of the kinetic model for the artificial metabolic pathways with the experimental results, whereby the differential equations were solved numerically. The calculated time evolution profiles for the synthesis of PANI-ES (red solid line in Fig. 2e, Supplementary Note 3-3(ii)) and of the membrane growth (red solid line in Fig. 3b, Supplementary Note 3-3(i)) agree well with the experimental results. It should be noted that the kinetic model not only describes the time dependence of the formation of the end products of the artificial metabolic pathways, i.e., the synthesis of the information polymer and the membrane growth, but it also successfully describes the consumption of the reactants (D-glucose and aniline) quantitatively during the reaction (Supplementary Note 4). The excellent agreement between the kinetic model and the experimental data indicates that the energy production, the synthesis of the information polymer, and the membrane growth units are integrated as designed, which is an indispensable element for achieving a synthetic minimal cell system.

**Recursive vesicle reproduction**. To construct a recursive vesicle reproduction system comprising artificial metabolic pathways, the vesicle membrane must grow to the limiting shape consisting of two spherical vesicles connected by a narrow neck. The offspring spherical vesicles are then produced by breaking the neck. The recursive vesicle reproduction cycle is completed after recovering the offsprings' surface area and volume to those of the mother vesicle. However, the AOT vesicles coupled with the synthesis of PANI-ES grow into a prolate shape rather than to the limiting shape, as shown in Fig. 3a. A key to attain vesicle growth to the limiting shape and vesicle division is the coupling between membrane curvature and molecular shape of the amphiphiles[12,27] (see **Discussion**). In our previous work[23], we introduced Chol having an inverse-cone molecular shape (small head and bulky tails) as a second component of the AOT GUVs. Binary AOT + Chol GUVs coupled with PANI-ES synthesis showed growth to the limiting shape vesicle and then vesicle division, i.e., production of daughter vesicles having almost the same size (symmetric division). The daughter vesicles, however, produced multiple smaller vesicles (asymmetric division) and no further reproduction was observed, i.e., no recursive vesicle reproduction occurred.

In the previous AOT + Chol/PANI-ES vesicle system[23], the initial vesicles contained Chol, which was not supplied during the reproduction process. Thus, the concentration of Chol in the vesicle membrane decreased from generation to generation, which might have been responsible for the observed cessation of vesicle division. To prevent the depletion of Chol, we supplied Chol to the vesicles by using sodium dodecylbenzenesulfonate (SDBS) + Chol mixed micelles, i.e., SDBS micelles containing solubilised Chol were added to the vesicle suspension. In these experiments, SDBS molecules having a sulfonate head group were also incorporated into the AOT membranes through interactions with PANI-ES, as in the case of AOT alone[23]. Therefore, SDBS + Chol mixed micelles played the role as transporter of Chol to the AOT vesicle membrane. In this study, both AOT micelles and SDBS + Chol mixed micelles were supplied to binary GUVs composed of AOT and Chol (9/1, mol ratio) under the optimised cascade reaction conditions (see **Methods**). The binary GUVs showed repeated vesicle membrane growth and division: as shown in Fig. 4a and Supplementary Movie 2, one of the mother GUV (#1) produced daughter GUVs (#2a and #2b, 26 sec), the daughter GUVs (#2a) produced granddaughter GUVs (#3a and #3b, 57 sec), and one of the granddaughter GUVs (#3b) produced great-granddaughter GUVs (#4a and #4b, 67 sec). We performed such membrane growth and division experiments 47 times. Similar vesicle reproductions up to the fourth generation were obtained 21 times (Fig. 4b). It should be noted that up to the third generation (#3a and #3b), the vesicles showed symmetric divisions, but these GUVs showed asymmetric division to produce fourth generation GUVs (#4a and #4b). This issue will be discussed in the **Discussion** section.

Despite the motivating results obtained, this vesicle reproduction system has one drawback. During the reproduction process, the total vesicle membrane area increases with time due to the incorporation of added AOT, SDBS, and Chol molecules, whereas the total vesicle volume remains nearly constant, as shown in Fig. 4c. Thus, to attain a recursive vesicle reproduction, the volume of the daughter GUVs must be recovered to that of the mother GUV. This could be achieved by osmotic swelling using vesicle-trapped osmolytes. When a vesicle encapsulates higher concentration of osmolytes than in a bulk solution, an inflow of water expands the vesicle volume. However, the osmotic pressure difference between the inside and outside of the vesicle decreases with time due to a dilution of the internal osmolytes, which eventually leads to the cessation of inflation. For example, when

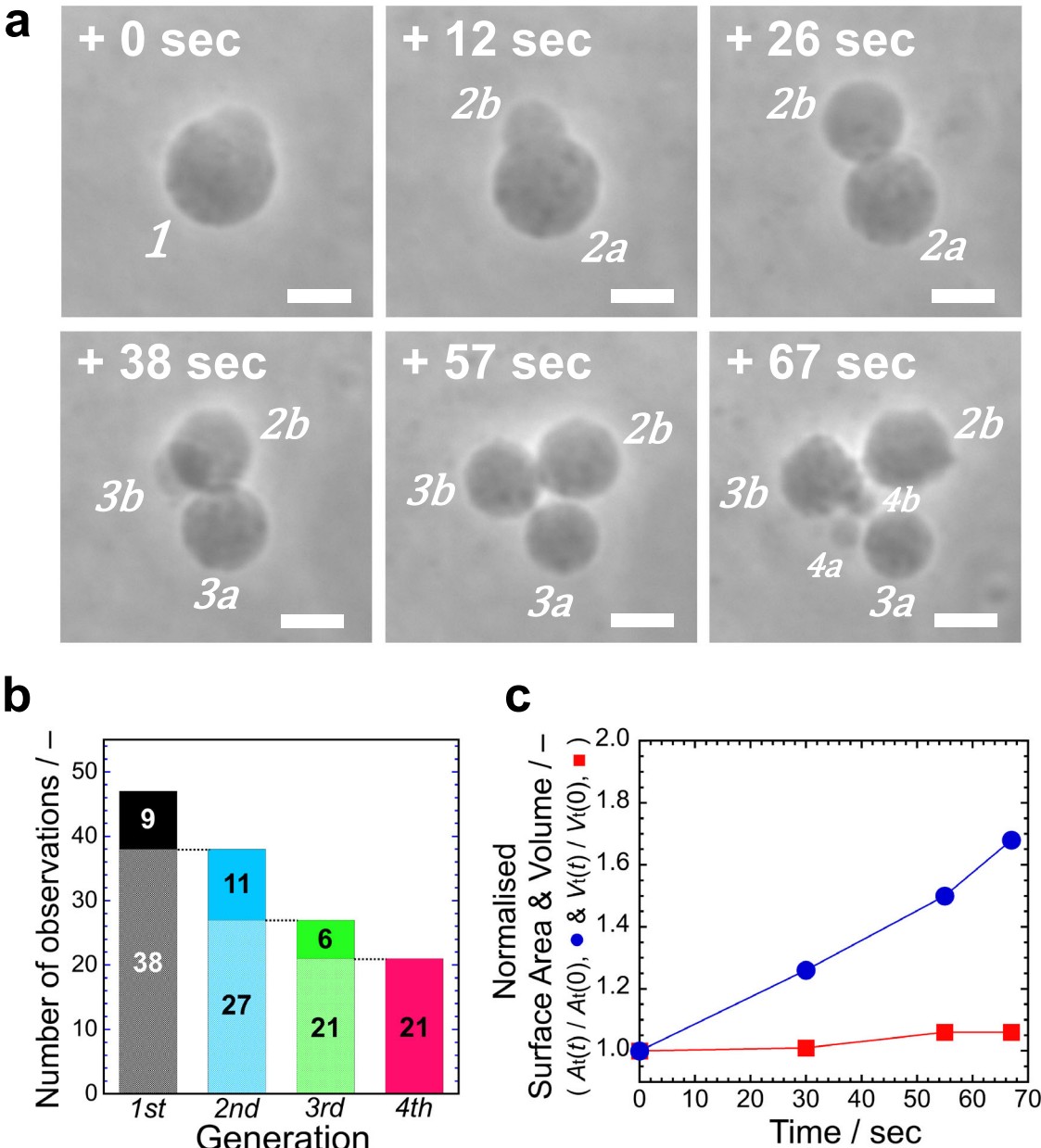

**Fig. 4 Membrane growth and division of binary AOT + Chol GUVs coupled with the synthesis of PANI-ES. a** Phase contrast light microscopy images of binary AOT + Chol (9/1, mol ratio) GUVs during the polymerisation reaction of aniline, supplied with D-glucose and micelles (AOT micelles and mixed SDBS + Chol micelles) (see **Methods**). The mother GUV (#1) produced two daughter GUVs (#2a and #2b), one of the daughter GUVs (#2a) divided into two granddaughter GUVs (#3a and #3b), and one of the granddaughter GUVs (#3b) produced small great-granddaughter GUVs (#4a and #4b). Complete division of the GUVs was confirmed by further observation. The elapsed time after starting the micro-injection is indicated in each image. Length of the scale bars: 10 μm. See also **Supplementary Movie 2**. **b** Statistics about the sustainability of the divisions of binary AOT + Chol (9/1) GUVs. Out of a total of 47 observations, the second-generation vesicles (#2a and #2b in **a**) were produced in 38 observations, third-generation vesicles (#3a and #3b in **a**) were produced in 27 observations, and fourth-generation vesicles (#4a and #4b in **a**) were produced in 21 observations. See also Supplementary Fig. 6a for the size distribution of the GUVs in each generation. **c**, Changes in normalised total surface area ($A_t(t)/A_t(0)$, blue circles) and total volume ($V_t(t)/V_t(0)$, red squares) of all GUVs originating from an initial mother GUV (#1 in **a**). The lines between the data points are drawn for guiding the eyes.

GUVs encapsulating a 100 mM D-sucrose solution were placed in a 90 mM D-sucrose solution containing 3.0 mM AOT, the volume increase reached the upper limit in ~80 sec due to dilution of the encapsulated osmolyte (*i.e.*, short-term swelling) (Supplementary Note 5 and Supplementary Fig. 5d). Therefore, long-term swelling requires maintaining the constant osmotic pressure difference between the inside and outside of the vesicle. One plausible way is to utilise two types of osmolytes having different membrane permeabilities. Here, GUVs encapsulating

100 mM D-sucrose solution were placed in a 100 mM D-fructose solution containing 3.0 mM AOT, whereby the AOT membrane is permeable to D-fructose, while the membrane permeability of D-sucrose is low. The coupling of D-fructose permeation into the interior of the vesicles with the subsequent flow of water into the vesicles maintains an osmotic pressure difference. The continuous volume expansion exerts an osmotic tension on the vesicle membrane. To release this membrane tension, AOT molecules in the external solution (critical vesiculation concentration; cvc

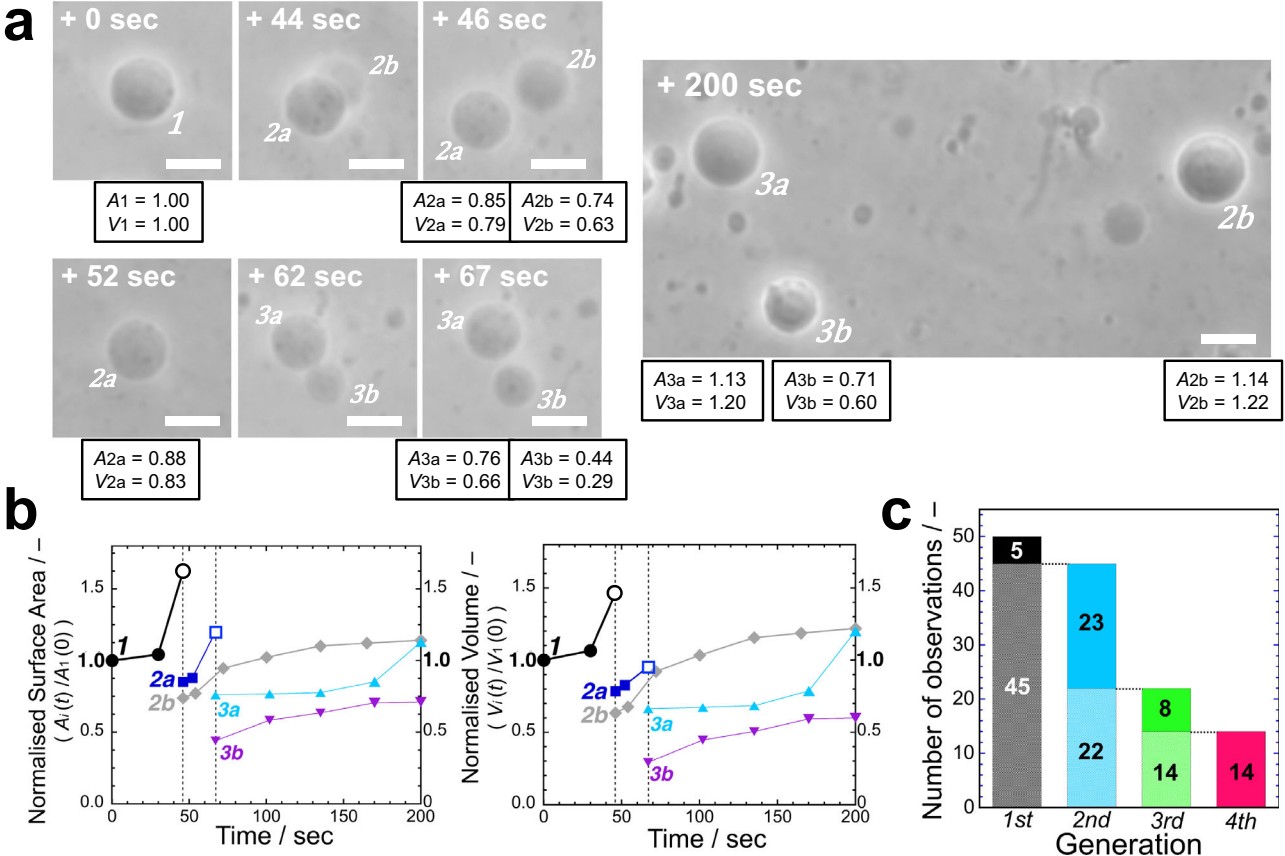

**Fig. 5 Recursive reproduction of the synthetic minimal cell system presented in this work. a** Phase contrast light microscopy images for the recursive reproduction of our synthetic minimal cell system caused by artificial metabolic pathways. The experimental protocol consists of the integrated membrane growth/division procedure that combines the vesicle surface-confined polymerisation of aniline (Fig.4a) and the sustainable osmotic vesicle swelling using the asymmetric permeation of osmolytes (Supplementary Fig. 5) (see **Methods**). The mother GUV (#1) produced two daughter GUVs (#2a and #2b). One of the daughter GUVs (#2a) produced two granddaughter GUVs (#3a and #3b), while another daughter GUV (#2b) was floated away. After the production of the granddaughter GUVs (#3a and #3b, ~70 sec), the external supply of D-glucose and micelles ceased to wait for the volume to recover. $A_i$ and $V_i$ ($i$ = 1, 2a, 2b, 3a, and 3b) in the microscopy images denote the surface area and volume of each offspring GUV normalised by the initial surface area and volume of the mother GUV (#1), $A_1(0)$ and $V_1(0)$, respectively, whereby we assume that the GUVs have axisymmetric shapes. In contrast to the vesicle reproduction system shown in Fig. 4, which lacks a volume inflation mechanism, after vesicle division the offspring GUVs grew to nearly the original surface area and volume of their mother GUV, i.e., $A_i/A_1(0)$ ~1.0 and $V_i/V_1(0)$ ~1.0. Length of the scale bars: 10 μm. See also Supplementary Movie 3. **b** Time evolutions of the normalised surface area ($A_i(t)/A_1(0)$, left) and the volume ($V_i(t)/V_1(0)$, right) of all GUVs originating from the same mother GUV(#1) in **a**. Vertical dashed lines represent the division events of the GUVs; #1 (black circles) was divided at 46 sec into #2a (blue squares) and #2b (grey diamonds), and #2a (blue squares) was divided at 67 sec into #3a (light blue triangles) and #3b (purple inverted triangles). The open black circle for GUV #1 and the open blue square for GUV #2a were obtained by summing the initial surface area or volume (on each dashed line) of the two GUVs they produced. It should be noted that the supply of D-glucose and membrane molecules ceased at ~70 sec, when the granddaughter GUVs (#3a and #3b) were produced. The total normalised surface area ($A_t(t)/A_1(0)$) and volume ($V_t(t)/V_1(0)$) of three offspring GUVs (#2b, #3a, and #3b) at $t$ = 200 sec reached values of 2.98 and 3.02, respectively. The lines between the data points are drawn for guiding the eyes. **c** Statistics about the sustainability of the recursive reproduction of binary AOT + Chol (9/1) GUVs. Out of 50 total observations, second-generation vesicles (#2a and #2b in **a**) were produced in a total of 45 cases, third-generation vesicles (#3a and #3b in **a**) were produced in 22 cases, and fourth-generation vesicles (smaller vesicles surrounding #3a in **a**) were produced in 14 cases. See also Supplementary Fig. 6b for the size distribution of the GUVs in each generation.

~1.5 mM) incorporate into the AOT membrane, resulting in an osmotic swelling of the vesicles. In the applied D-sucrose/D-fructose system, the volume of AOT GUVs increased almost linearly with time up to about 1.7 times during 700 sec (Supplementary Fig. 5d), which indicates that the asymmetric osmolytes strategy is effective for achieving long-term swelling.

Finally, we integrated all the procedures to attain a recursive reproduction of the synthetic minimal cell system. Binary AOT + Chol (9/1) GUVs encapsulating 100 mM D-sucrose were placed in a solution containing isomolar D-fructose but being otherwise identical with the optimal cascade reaction mixture used for the artificial metabolic pathways. Then, D-glucose and both AOT micelles and SDBS + Chol mixed micelles were

supplied to a target AOT + Chol GUV using the double micro-injection technique (see **Methods**). This resulted in the complete reproduction of the target vesicle, i.e., membrane growth, vesicle deformation, division, and volume recovery (Fig. 5a and Supplementary Movie 3). When the cascade reaction proceeded by supplying D-glucose and vesicle-forming molecules, the mother AOT + Chol GUV (#1) showed membrane growth and division to produce two daughter GUVs (#2a and #2b) at 46 sec, and one of the daughter GUVs (#2a) produced granddaughter GUVs (#3a and #3b) at 67 sec. After the production of these granddaughter GUVs, the external supply of vesicle-forming molecules ceased at ~70 sec, and then the offspring GUVs recovered their volume nearly to their mother's volume. All in all,

the mother binary GUV reproduced to yield daughter (#2b) and granddaughter GUVs (#3a) with almost the same surface area and volume in 200 sec, *i.e.*, recursive reproduction of our synthetic minimal cell system was achieved successfully (Fig. 5b). Similar experiments were performed 50 times. In 14 cases, recursive vesicle reproduction was observed up to the fourth generation. (Fig. 5c, Supplementary Note 6 and Supplementary Movie 4–6).

## Discussion

In this study, we demonstrated recursive vesicle reproduction maintained by chemical transformations from D-glucose to PANI-ES. Obviously, our synthetic minimal cell system consists of pathways that are completely different from those of living systems. Nevertheless, it is an interesting vesicular compartment system, in which vesicle membranes grow by directly incorporating externally supplied membrane molecules (Fig. 1b) through specific interactions with the information polymer PANI-ES. This simplification in our model system allowed us to describe the artificial metabolic pathways quantitatively, see Eqs. (6)–(16). The reaction steps summarised in Fig. 1b show that our synthetic minimal cell has the following information flow. The AOT membrane (genotype) is encoded in the *para*-NC sequence of PANI-ES (phenotype), which promotes AOT vesicle reproduction (fitness).

In the following, we will first discuss the physical background of the information flow. In the artificial metabolic pathway, the aniline radical cation ($S^*$), can be described with four resonance structures to illustrate the delocalisation of the unpaired electron, *i.e.*, at the nitrogen atom (N), at the carbon atom in *ortho*-position (two positions; O1 and O2), or in the *para*-position (P) (R3 in Fig. 1b)[25]. On the AOT membrane, dimerisation of $S^*$ and further elongation reactions occur preferentially between the carbon atom in the *para*-position to the amino group and the nitrogen atom (linear *para*-NC sequence in PANI-ES) due to specific interactions between the aniline radical cation and AOT (template effect) (R4 in Fig. 1b)[25,34]. Another role of the specific interactions is the binding of AOT molecules present in the external solution to PANI-ES, which reduces the hydrophilicity of the bound AOT molecules and promotes the incorporation into the AOT membrane, *i.e.*, resulting in the growth of the membrane (R9 in Fig. 1b). Without such specific interactions, aniline radical cation coupling occurs randomly, resulting in the formation of random sequence polyaniline[34,35,45], with which no membrane growth is observed[23]. Thus, the sequence in PANI determines the ability to promote membrane growth.

The heart of our synthetic minimal cell is the coupling between the synthesis of an information polymer (PANI-ES) and vesicle reproduction, which originates in specific interactions between the aniline radical cation and the sulfonate head group of AOT. This resembles the Watson-Crick base pairing in the proliferation scheme of living cells. Here, we identify the thermodynamic conditions for the stable encoding of the specific sequences of the polymer. The sequence information is quantified by the Shannon entropy[46] given by $H(l) = -\sum_{S^*} f_{S^*,l} \log_2 f_{S^*,l}$ (bit per monomer), where $f_{S^*,l}$ is the probability of having four aniline radical cations $S^* \in \{N, O1, O2, P\}$ at position $l$ in polyaniline. For a random sequence, *i.e.*, $f_{S^*,l} = 1/4$, the Shannon entropy is $H_r(l) = 2$ (bit per monomer)[47]. Then, the information at position $l$ is expressed by the difference between the random sequence and the specific sequence, $R(l) = H_r(l) - H(l) = 2 - H(l)$. If the polyaniline has a completely ordered *para*-NC sequence, the information of PANI-ES at position $l$ is $R^{ES}(l) = 2$ (bits). The thermodynamic entropy loss due to the formation of such regular polyaniline segment composed of four monomers (PANI-ES segment, R9 in Fig. 1b) is

given by $\Delta S = -k_B \ln(2) \sum_{l=1}^{4} R^{ES}(l) = -8\ln(2)k_B$ (J K$^{-1}$ per segment). The second law of thermodynamics requires the inequality $\Delta H - T\Delta S \leq 0$, where $\Delta H$ is the enthalpy change. To synthesise PANI-ES having the specific sequence information, this entropy loss should be compensated by the enthalpy gain from the template effect. Considering one repeating unit of PANI-ES with two amines and two amine radical cation groups (*i.e.*, half-oxidised and half-reduced tetraaniline repeating unit) (R9 in Fig. 1b), PANI-ES is obtained as a complex involving template amphiphiles as counterions, such as AOT, stabilised by specific interactions[25,31,48,49] (see Supplementary Fig. 1a). For the formation of the PANI-ES–AOT complex, the enthalpy change, which is mainly due to the formation of specific hydrogen bonds (electrostatic attraction-involved hydrogen bonding)[50], is estimated as $\Delta H \sim -103k_B T$ ($<T\Delta S = -8\ln(2)k_B T$) per PANI-ES segment (*i.e.*, tetramer) (Supplementary Note 7). Thus, the formation of the *para*-NC sequence in PANI-ES is stabilised by the bond energy between PANI-ES and AOT.

It should be noted that PANI-ES, which encodes in its segment sequence the AOT membrane, plays a role as catalyst for promoting AOT vesicle growth. The entire process can be viewed as being analogous to the essence of living systems, *i.e.*, the central dogma of molecular biology, where DNA with its nucleic acid sequence information (genotype) is transferred to the amino acid sequence of enzymes (phenotype). The enzymes catalyse metabolic reactions and lead to cell proliferation (fitness). The important conceptual difference between our synthetic minimal cell system and living systems is the feedback of information. In our synthetic minimal cell, the AOT membrane (genotype) determines the sequence of PANI-ES (phenotype), and PANI-ES promotes the selective growth of the AOT membrane, in contrast to the central dogma where no information is transferred from proteins to nucleic acids[51].

Another essence of our synthetic minimal cell system is the recursive vesicle reproduction, *i.e.*, transformations that occur along the following steps: spherical vesicle → growth to limiting shape vesicle → vesicle division → recovery to original spherical vesicle size (Fig. 5a). In the following, we will discuss the membrane physics background of the recursive vesicle reproduction. The first step, spherical vesicle → growth to limiting shape vesicle, starts by incorporating membrane molecules into the vesicle. According to Fick's law, the flux of membrane molecules from the external solution to the membrane, $J$, is related to the change in the chemical potential by $J \propto (\mu_{ext} - \mu_{ves})$, where $\mu_{ext}$ and $\mu_{ves}$ are the chemical potentials of a membrane molecule (AOT and SDBS) in the external solution adjacent to the vesicle and in the vesicle membrane, respectively[12]. Due to specific interactions, PANI-ES modifies the chemical potential of AOT in the external solution, $\mu_{ext}$ (decrease in hydrophilicity of the bound AOT (SDBS) molecules; Supplementary Fig. 1), and promotes the incorporation of AOT and SDBS (together with Chol) into the outer leaflet of the vesicle membrane. Excess membrane molecules in the outer leaflet are transported to the inner leaflet by flip-flop motions[52], which results in the growth of the ternary AOT + SDBS + Chol vesicle to the limiting shape vesicle (Fig. 5a).

This growth to the limiting shape vesicle is described as follows. For simplicity, we consider a binary AOT + Chol vesicle that shows the same growth process as the ternary AOT + SDBS + Chol vesicle. The deformation of the AOT + Chol vesicle is described by the spontaneous curvature model[29,53] because AOT and Chol have fast flip-flop rates[52,54]. The spontaneous curvature model shows that the vesicle shape is determined by the reduced volume $v = V/[(\frac{4\pi}{3})R_0^3]$ ($R_0 = \sqrt{A(t)/4\pi}$, $V$; vesicle volume and $A$; vesicle surface area) and the reduced spontaneous curvature $c_0 = C_0 R_0$ ($C_0$; spontaneous curvature)[55]. For a binary

AOT + Chol vesicle, the reduced spontaneous curvature of the membrane is expressed by $c_0 = \frac{1}{2} H_{Chol} \triangle \phi R_0$, where, $H_{Chol}$ is the molecular spontaneous curvature of Chol and $\triangle \phi = \phi^+ - \phi^-$ ($\phi^+$ and $\phi^-$ are the area fraction of Chol in the outer and inner leaflet, respectively)[28]. According to the phase diagram based on the spontaneous curvature model[55], the symmetric limiting shape vesicle is obtained at $v = 0.7$ and $c_0 = 3$, and $\Delta \phi$ required to attain $c_0 = 3$ is only $\Delta \phi \sim -2 \times 10^{-3}$ (Supplementary Note 8-1). When $c_0$ is larger than 3, the vesicle deforms to an asymmetric limiting shape (formation of large and small daughter vesicles). On the contrary, when $c_0$ is smaller than 3, the vesicle cannot deform into the limiting shape. In the membrane growth stage of our synthetic minimal cell system, AOT molecules present in the external solution are incorporated into the outer leaflet of the vesicle bilayer through PANI-ES and then move to the inner leaflet by flip-flop motions. Therefore, $\triangle \phi$ is determined by a balance between AOT uptake rate and AOT flip-flop rate. If the AOT uptake rate is faster than the AOT flip-flop rate, the concentration of Chol in the outer leaflet is diluted, which results in increase of $\Delta \phi$. Thus, the AOT uptake rate determined by PANI-ES might be responsible for the deformation of AOT + Chol/PANI-ES vesicle to the limiting shape[23].

To attain spontaneous vesicle division, it is necessary that (1) the two-vesicle state after division is more stable than the limiting shape state before division, and (2) the energy barrier between the two states – limiting shape and two separate vesicles—can be overcome with the thermal energy. The free energy difference between the limiting shape state, $F_1$, and the two-vesicle state (two vesicles having the same radius $R_t$), $F_2$, is expressed by $F_2 - F_1 \cong -4\pi a_{ne}\kappa \left[ C_0 - \frac{2}{R_t} \right] + 4\pi\kappa_G$, where $a_{ne}$ is the neck radius ($a_{ne} \ll R_t$), $\kappa$ is the bending rigidity, and $\kappa_G$ is the Gaussian bending rigidity[12,56]. Although the estimation of the Gaussian bending rigidity, $\kappa_G$, is still controversial[57,58], we use a rough estimation, $\kappa_G \sim -\kappa$[59]. For the symmetric limiting shape vesicle with $c_0 = 3$, we obtain $F_2 - F_1 \sim -4\pi\kappa < 0$. Thus, the two-vesicle state is more stable than the limiting shape state, i.e., the energy barrier is important here. The key to achieve vesicle division for binary vesicles is the coupling between Gaussian curvature and local lipid composition (AOT and Chol) through their Gaussian curvature rigidities[12,26,27,60]. Since the neck of the limiting shape vesicle has a negative Gaussian curvature, the amphiphile having a larger Gaussian curvature rigidity prefers to localise in the neck region, whereas the amphiphile having a smaller Gaussian curvature rigidity prefers to localise in the spherical segment. Here, we consider a binary AOT + 1,2-dilauroyl-sn-glycero-3-phosphoethanolamine (DLPE) vesicle that exhibits vesicle growth and division similar to ternary AOT + SDBS + Chol vesicles (Supplementary Note 8-2), since DLPE has an inverse-cone shape. The reported bending rigidities for AOT and DLPE are $\kappa^{AOT} \sim 3$ $k_B T$[61] and $\kappa^{DLPE} \sim 39$ $k_B T$[62], respectively. Then, we estimate $\kappa_G^{AOT} \sim -3$ $k_B T$ and $\kappa_G^{DLPE} \sim -39$ $k_B T$, i.e., $\kappa_G^{DLPE} < \kappa_G^{AOT}$. The large difference in Gaussian bending rigidities between AOT and DLPE causes the segregation of AOT and DLPE at the neck region, which produces an interface between the neck region and the spherical cap region. This interface might reduce the energy barrier and destabilises the neck, causing the spontaneous vesicle division, as shown in Supplementary Fig. 8. Our coarse-grained molecular dynamics simulation study clearly shows that the inverse cone shaped lipid induces the spontaneous vesicle division[60]. We suppose that this scenario is also valid for AOT + SDBS + Chol/PANI-ES systems (Figs. 4a and 5a).

The interplay between the permeable osmolyte (D-fructose) present in the external solution and the less permeable osmolyte (D-sucrose) encapsulated in the vesicles enables a long-term vesicle volume increase. The volume increase of AOT GUVs

caused by the asymmetric permeation of the two osmolytes is expressed by

$$\frac{dn_f(t)}{dt} = P_f A(t) \left( c_{ext} - \frac{n_f(t)}{V(t)} \right) \qquad (17)$$

$$\frac{dV(t)}{dt} = v_w \frac{dn_w(t)}{dt} = P_w A(t) v_w \left( \frac{n_f(t) + n_s}{V(t)} - c_{ext} \right), \qquad (18)$$

where we assume that AOT molecules in the external solution are instantly incorporated into the vesicle membrane to relax the membrane tension; $n_f, n_w$, and $n_s$ are the amount of D-fructose, water, and D-sucrose molecules encapsulated in the AOT GUV, respectively. $P_f$ and $P_w$ are the permeabilities of D-fructose and water against the AOT membrane, $c_{ext}$ is the concentration of D-fructose in the external solution, and $v_w$ is the molar volume of water molecules. The simultaneous differential equations Eqs. (17) and (18) describe the observed growth profiles well (Supplementary Note 5). All in all, recursive vesicle reproduction is achieved by integrating the mentioned membrane growth, deformation, division, and volume recovery processes.

To sustain the recursive vesicle reproduction, it is necessary to regulate the reduced spontaneous curvature, $c_0$, in the deformation process to be ~3 (see above). A key to regulate $c_0$ is the balance between the uptake rate and the flip-flop rate of AOT. When the uptake rate is larger than the flip-flop rate, $c_0$ increases gradually with time, which results in a change from symmetric to asymmetric vesicle division, as shown in Fig. 4a at 67 sec[55]. Thus, to attain recursive reproduction of our synthetic minimal cell over several generations (Fig. 5a), we regulated the polymerisation rate for PANI-ES formation and the supply rate of AOT micelles and mixed SDBS + Chol micelles by adjusting the micro-injection pressure manually. However, even if we regulated the injection pressure, the vesicle divisions became generally asymmetric with generations (Supplementary Note 6), whereby some daughter vesicles recovered their mother vesicle size in the volume inflation stage (see also Fig. 5b). It should be noted, however, that it requires further elaborations of the experimental setup to systematically control the reduced volume $v$ and $c_0$ to attain sustainable symmetric vesicle division.

Based on the above discussion, in our minimal cell, the membrane molecules must satisfy the following three requirements: (1) vesicle membrane growth coupled with an information molecule, (2) deformation of the initial vesicle to the limiting shape vesicle and division of the limiting shape vesicle, and (3) inflation of the volume of the obtained vesicles to the volume of the initial vesicle. Condition 1 requires that the vesicle-forming amphiphile must have a sulfonate head group (R-SO$_3^-$). A high cvc is also required for amphiphiles to be incorporated into the vesicle membrane. Condition 2 requires the introduction of inverse cone-shaped membrane molecules as the second component of the membrane. In addition, the balance between the uptake rate and the flip-flop rate of the membrane molecules determines the deformation to the limiting shape vesicle. Condition 3 requires regulating the permeabilities of water and osmolytes by adjusting the structure of the hydrophobic part of the membrane molecules.

Finally, we compare the reproduction of our synthetic minimal cell with the division mode of bacterial L-form cells (called L-forms) as simple proliferation model organisms[63–65]. While the cell division of bacteria is generally regulated by the Z-ring, i.e., ring-shaped filaments of the FtsZ protein[66], the division mode of the L-forms is not mediated by a Z-ring, which is considered a model for proliferation in primitive cells. Although the reproduction of our synthetic minimal cells is achieved by the uptake of externally added membrane molecules (and not by in situ lipid synthesis), we find several similarities between the reproduction

of our synthetic minimal cells and the proliferation of L-forms: (i) the production of small daughter cells, tubulation, vesiculation and re-fusion of daughter cells observed in the L-form proliferation process are frequently observed in experiments with our synthetic minimal cells, (ii) the synthesis of certain branched-chain fatty acids is critical for the L-form proliferation, which agrees well with the importance of the inverse-cone shaped lipids in the reproduction of our synthetic minimal cell[67], and (iii) volume inflation attained by osmoregulatory proteins (in bacteria)[68,69] is similar to the swelling of our synthetic minimal cell system, where an asymmetric distribution of osmolytes controls water permeation. The great advantage of the synthetic minimal cell in terms of the observed cell compartment transformations is that the physical background of the elementary processes is clear, as outlined above. Thus, understanding such artificial reproduction mechanisms might be useful not only for developing synthetic minimal cells, but possibly also for discovering the reproduction mechanism of primitive, i.e., very early cells.

Our synthetic minimal cell reproduces itself through chemical and physical transformation networks composed of an energy production unit, an information polymer synthesis unit, and a vesicle reproduction unit (metabolism), leading to offsprings. By regulating the vesicle composition, the osmolytes, and the supply of ingredients (chemical flows), the offsprings show further reproductions (recursive reproduction). This reproduction process is based on an information flow, whereby the AOT membrane (genotype) is transferred to the para-NC coupled sequence of PANI-ES (phenotype) through specific interactions, which promotes the reproduction of AOT vesicles (fitness). Although in our synthetic minimal cell the chemical flows are artificially controlled, coupling chemical flows and an information flow is a great step toward the eventual formation of living systems in which the central dogma of molecular biology autonomously regulates them. Synthesising minimal cells is a promising approach to bridge the gap between nonliving forms of matter and living systems.

## Methods

**Chemicals**. AOT (sodium bis-(2-ethylhexyl) sulfosuccinate, >99%, catalogue No. 86139) was purchased from Sigma-Aldrich Japan (Tokyo, Japan). SDBS (sodium dodecylbenzenesulfonate, hard type (mixture), >95%, No. D0990) was purchased from Tokyo Chemical Industry (Tokyo, Japan). DOPC (1,2-dioleoyl-sn-glycero-3-phosphocholine, >99%, No. 850375), DLPE (1,2-dilauroyl-sn-glycero-3-phosphoethanolamine, >99%, No. 850702), and cholesterol (Chol, ovine wool, >98%, No. 700000) were purchased from Avanti Polar Lipids, Inc. (AL, USA). The amphiphiles were used without further purification and dissolved in chloroform at 100 mM (AOT and SDBS) or at 10 mM (others) and stored at –20 °C as stock solutions.

Aniline (>99%), hydrogen peroxide (30% in water, ~9.8 M), sodium dihydrogenphosphate (NaH$_2$PO$_4$) dihydrate (>99.0%), phosphoric acid (H$_3$PO$_4$, >85%), D(+)-glucose (>99%), D(-)-fructose (>99%), D-sucrose (>99%), and chloroform (>99%) were purchased from Wako Pure Chemical Industries (Osaka, Japan). HRPC (horseradish peroxidase isoenzyme C, Grade I, PEO-131, 286 U mg$^{-1}$, RZ = 3.13, Lot No. 74590) and GOD (glucose oxidase from *Aspergillus* sp., Grade II, GLO-201, 166 U mg$^{-1}$, Lot No. 74180) were purchased from Toyobo Enzymes (Osaka, Japan). All other chemicals used were of research-grade. The concentration of HRPC and GOD was determined spectrophotometrically using $\varepsilon_{403}$(HRPC) = 1.02 × 10$^5$ M$^{-1}$cm$^{-1}$,[70] and $\varepsilon_{450}$(GOD) = 2.82 × 10$^4$ M$^{-1}$cm$^{-1}$,[71] as molar absorption coefficients. Ultrapure water purified with a Direct-Q 3 UV apparatus (Millipore, USA) was used to prepare all aqueous solutions and suspensions.

**Preparation of GUVs, LUVs, and micellar solutions**. GUVs composed of AOT were prepared using the gentle hydration method[23,72]. First, AOT (17.8 mg) was dissolved in 1 mL chloroform in a 5 mL glass vial, followed by forming a thin AOT film upon removing chloroform with a rotary evaporator. Then, for complete removal of chloroform, the AOT film was put under a high vacuum overnight, while the vial was kept wrapped with aluminium foil. The dried AOT film was hydrated and dispersed at 60 °C for 1–2 h with 2.0 mL of 20 mM NaH$_2$PO$_4$ solution (pH = 4.3) or with a 2.0 mL of 20 mM NaH$_2$PO$_4$ solution containing

100 mM of D-glucose, D-fructose, or D-sucrose (pH = 4.3). This resulted in the formation of GUVs with radii of 5–30 μm. The obtained 20 mM AOT GUV suspension was stored at $T$ ~25 °C and used within two days after preparation.

Binary GUVs of AOT and Chol (9/1, molar ratio, 5.0 mM in total, 0.50 mL suspension) were prepared in the same way as described above, using a chloroform solution containing the two amphiphiles in the prescribed molar ratio. The only difference was that an AOT + Chol dried film was hydrated at 60 °C for 15–20 min.

A suspension of LUVs composed of AOT or DOPC with an average diameter of ~80–100 nm were prepared by using the freezing-thawing extrusion method[23,25,30]. First, GUV suspensions were prepared as described above, and the obtained suspensions were frozen in liquid nitrogen and thawed in a 60 °C water bath. This operation was repeated ten times. Then, the suspensions were extruded ten times through a 200 nm pore-size nucleopore polycarbonate membrane, and another ten times through a 100 nm pore-size nucleopore membrane by using the LIPEX$^{TM}$ Extruder (Northern Lipids Inc., Canada). The LUVs suspensions were stored at $T$ ~25 °C and used within 7 days.

A 20 mM AOT micellar solution containing 100 mM D-glucose was prepared by simply dissolving solid AOT in a 100 mM D-glucose solution, followed by mixing with the help of a vortex mixer. It should be noted that AOT molecules do not form vesicles but form micelles in the absence of NaH$_2$PO$_4$. 100 mM SDBS + 0.5 mM Chol mixed micelles were prepared from the stock solutions in chloroform. First, 500 μL of 100 mM SDBS solution in chloroform and 25 μL of 10 mM Chol solution in chloroform were mixed in a 5 mL glass vial. The chloroform was removed using a nitrogen gas stream, and the vial wrapped with aluminium foil was put under a high vacuum overnight. Then, the dried SDBS + Chol mixture was hydrated with 0.5 mL of 100 mM D-glucose solution using a vortex mixer, which resulted in the formation of a SDBS + Chol (100 mM/0.5 mM) mixed micellar solution containing 100 mM D-glucose. It should be noted that if the molar ratio of Chol to SDBS was further increased, we observed precipitation of Chol in the SDBS micellar solution within a few days.

**Optimised cascade reaction conditions for the polymerisation of aniline in the presence of AOT LUVs inside reaction tubes**. The enzymatic cascade polymerisation of aniline in the presence of AOT LUVs as templates was carried out in 5 mL Eppendorf polypropylene tubes using the following protocol, based on our previous work[30]. All components of the reaction mixture, except the GOD solution, were added to 216.7 μL of a 20 mM NaH$_2$PO$_4$ solution (pH = 4.3) containing 100 mM D-glucose: 75 μL of the prepared LUV suspension (20 mM AOT in 20 mM NaH$_2$PO$_4$ solution containing 100 mM D-glucose), 50 μL of an aqueous aniline solution (40 mM in 20 mM NaH$_2$PO$_4$ solution, pH adjusted to 4.3 with H$_3$PO$_4$), 25 μL of a HRPC solution (18.4 μM in 20 mM NaH$_2$PO$_4$ solution, pH = 4.3), and 83.3 μL D-glucose solution (250 mM in 20 mM NaH$_2$PO$_4$ solution, pH = 4.3). After gentle mixing, the reaction was triggered by the quick addition of 50 μL of a GOD solution (10 μM in 20 mM NaH$_2$PO$_4$ solution, pH = 4.3), followed by gentle mixing, closing of the lids of the reaction tubes, and sealing with Parafilm. The tubes were then placed in a rotary mixer Magic Mixer TMM with an LH15x22 rack (KENIS, Japan) and continuously rotated at ~90 rpm during the reaction. The initial reaction conditions were as follows: 3.0 mM AOT, 4.0 mM aniline, 0.92 μM HRPC, 1.0 μM GOD, 100 mM D-glucose, and dissolved oxygen in 20 mM NaH$_2$PO$_4$ solution (pH = 4.3), reaction volume = 0.50 mL, $T$ ~25˚C. For reference, DOPC LUVs and AOT GUVs were used instead of AOT LUVs.

**Cascade reaction conditions for the polymerisation of aniline in the presence of GUVs using the micro-injection technique**. First, 2.0 mL of the prepared AOT or AOT + Chol (9/1, molar ratio) GUV suspension, to which all polymerisation components (except D-glucose) were added, were carefully transferred at room temperature ($T$ ~25 °C) from the Eppendorf tube into a sample chamber, a glass-bottom dish D11130H (Matsunami, Japan). The initial concentrations of each component of the reaction mixture were the same as the ones used for running the cascade reaction in the presence of LUVs (with the exception of D-glucose): 3.0 mM amphiphiles (GUVs), 4.0 mM aniline, 0.92 μM HRPC, 1.0 μM GOD, and dissolved oxygen in 20 mM NaH$_2$PO$_4$ solution (pH = 4.3). The polymerisation was triggered by micro-injecting a 100 mM D-glucose solution containing micelles (20 mM AOT and/or 100 mM SDBS + 0.5 mM Chol) by using a double micro-injection technique with a symmetric configuration of two micro-pipettes, as described previously[23]. One of the two micro-pipettes were used for the micro-injection of the D-glucose-containing suspension or solution. Unless specifically mentioned otherwise, the same solution as the bulk solution (but without GUVs and D-glucose) was injected from the other micro-pipette as a counter flow: 4.0 mM aniline, 0.92 μM HRPC, and 1.0 μM GOD in 20 mM NaH$_2$PO$_4$ solution (pH = 4.3). All solutions injected from the micro-pipettes were pressed through a 0.2 μm polypropylene filter Puradisc 25 PP (GE Healthcare, UK) before use.

It should be noted that the GOD solution was used to trigger the polymerisation reaction in the presence of LUVs, while the D-glucose solution was used in all subsequent experiments with GUVs by the double micro-injection technique. This was to prevent an undesirable osmotic shock to LUVs. The addition of a large amount of D-glucose solution (>100 mM, >100 μL) to a glucose-free mixture containing LUVs (~0.5 mL) would have caused a non-negligible osmotic pressure difference across the LUV membranes (~100 mOsm L$^{-1}$). Thus, we prepared LUVs in a solution containing 100 mM D-glucose and triggered the reaction by adding

the GOD solution instead. This is just a replacement of the order of adding the reaction components, and there is no difference in terms of reaction kinetics of $H_2O_2$ production.

The micro-injection experiments were performed under a phase-contrast light microscope (see below) equipped with a hydraulic micro-manipulator MMO-202ND (Narishige, Japan), a Femtojet system, and a Femtotip II (Eppendorf, Germany) with a diameter of $0.5 \pm 0.2$ μm. The distance between the tips of the two micro-pipettes was ~100 μm, and the distance from the tip to the bottom of the chamber was ~40 μm. The injection pressure of the micro-pipettes was ~70 hPa, corresponding to ~0.10 nL sec$^{-1}$. The two injection flows trapped the GUV at an almost fixed position at the bottom of the chamber, which kept the distance between the tips of the micro-pipettes and the target GUV during the observations constant (for a schematic diagram, see also Supplementary Fig. 2-2a).

### Characterisation of polyaniline

*UV/Vis/NIR spectroscopy.* Absorption measurements in the UV/Vis/NIR region of the spectrum were carried out with a V-730 spectrometer (JASCO, Japan) at $T$ ~25 ℃, using quartz cuvettes S15-UV-1 (GL Sciences Inc., Japan) with an optical path length of $L = 0.1$ cm[23,25,30].

*Micro-Raman spectroscopy.* Raman spectra were obtained by using an inVia QONTOR confocal Raman spectrometer (Renishaw, UK), equipped with a diode-pumped solid-state laser (532 nm, 50 mW), an optical microscope, and a CCD detector. The Raman spectra of the reaction mixtures obtained in the presence of LUVs under the optimised cascade reaction conditions were collected in non-confocal mode with an objective (N Plan L50x, NA = 0.50 (Leica, Germany)). The reaction mixtures were placed in a holed silicone rubber sheet on a borosilicate glass slide. The hole had a diameter of 12 mm and a depth of 1 mm. The exposure time for one measurement run was 1.0 sec; 30 runs were accumulated with ~15 mW laser power on the sample stage. Raman mapping of PANI-ES on an AOT GUV using the characteristic $\nu(C{\sim}N^{\bullet+})_p$ peak at ~1345 cm$^{-1}$ was performed in confocal mode by using a water immersion objective (C-Apochromat 100x, NA = 1.25 (Carl Zeiss, Germany)). The laser spot size was 520 nm in diameter and 680 nm in depth. First, the reaction mixtures containing AOT GUVs were transferred into VitroTubes (#5001) (borosilicate rectangle glass tubes with a path length of 10 μm) (VitroCom, USA). Then both ends of the tubes were sealed with Capillary Wax (HR4-328) (Hampton Research, USA). The laser was focused on an immobilised GUV with a diameter of ~10 μm. The target GUV was horizontally translated by a piezoelectric stage with 0.5 μm steps in both $x$ and $y$ directions, and ~680 points Raman spectra were collected in total. The exposure time was 30 sec per point with ~1.5 mW laser power on the sample stage, and it took ~6 h to complete the entire scanning of the target AOT GUV. All obtained spectra were fluorescence corrected with the WiRE software (Renishaw, UK).

### Protocol for the reproduction of GUVs

The recursive reproduction of our vesicular synthetic minimal cell system, consisting of artificial metabolic pathways, was demonstrated using the micro-injection setup described above under osmotic swelling condition: vesicle membrane growth → vesicle deformation → vesicle division → vesicle inflation (Fig. 5a). First, AOT + Chol (9/1, 5 mM amphiphiles) binary GUVs were prepared in 20 mM NaH$_2$PO$_4$ solution (pH = 4.3) containing 100 mM D-sucrose (osmolyte) by using the gentle hydration method. Then, the polymerisation components (75 μL of a 267 mM D-sucrose solution, 50 μL of a 40 mM aniline solution, 25 μL of a 18.4 μM HRPC solution, and 50 μL of a 10 μM GOD solution; all solutions being prepared in 20 mM NaH$_2$PO$_4$ solution, pH = 4.3) were added to 300 μL of the prepared AOT + Chol GUV suspension in a 2.0 mL polypropylene tube. This mixture contained 3.0 mM amphiphiles (AOT + Chol), 4.0 mM aniline, 0.92 μM HRPC, 1.0 μM GOD, dissolved oxygen, 100 mM D-sucrose, and 20 mM NaH$_2$PO$_4$ (pH = 4.3). After gentle mixing, 120 μL of the mixture containing GUVs was dropped into the hole in the silicone sheet chamber for the light microscope observation. Then, a small volume containing a selected GUV was carefully transferred into 2.0 mL of a 20 mM NaH$_2$PO$_4$ solution (pH = 4.3) containing 3.0 mM AOT, 4.0 mM aniline, 0.92 μM HRPC, 1.0 μM GOD, dissolved oxygen, and 100 mM D-fructose, which was previously added to the glass bottom chamber. It should be noted that when the membrane growth and division of the GUVs were demonstrated without the volume recovery (Fig. 4a), this transfer procedure was skipped, and the micellar solution containing D-glucose was supplied as described below to the selected GUVs located in the holed silicone chamber. For the transfer of the GUV, a transfer pipette VacuTipII and Cell Tram Vario (Eppendorf, Germany) was used. The bulk solution that was placed in the glass bottom chamber was passed through a 0.2 μm polypropylene filter Puradisc 25 PP (GE Healthcare, UK) before use. The difference in components between inside and outside the AOT + Chol (9/1) GUV was the osmolytes (100 mM D-sucrose inside the GUV and 100 mM D-fructose outside the GUV). Due to the differences in molecular structure, D-fructose can penetrate across the vesicle membrane from the external solution into the aqueous interior of the vesicles, whereas the vesicle membrane permeability for D-sucrose is low (Supplementary Note 5). Then, the permeation of D-fructose couples with the osmotic drag of water, causing long-term swelling, while AOT molecules present in the external solution are incorporated into the membrane to release the surface tension generated by the swelling. Quickly after the transfer of the AOT + Chol (9/1) GUV,

the polymerisation was triggered by the micro-injection of a 100 mM D-glucose solution containing micelles (20 mM AOT and 100 mM SDBS + 0.5 mM Chol). The configuration and injection pressure were the same as described above. After the production of granddaughter GUVs (#3a and #3b in Fig. 5a) at ~70 sec, the micro-injection was stopped to suppress membrane growth and to recover the vesicle volume.

### Microscope observation of GUVs

The morphological changes of GUVs were followed by using an Axio Vert A1 FL-LED inverted fluorescence microscope in phase-contrast mode (Carl Zeiss, Germany) with a dry 40x objective (LD A-Plan 40x, NA = 0.55) and a CCD camera (Axiocam 506mono) (Carl Zeiss, Germany) for recording the images. To estimate the vesicle surface area and volume quantitatively, a 3D image of the GUV was reconstructed from the 2D microscope image by using the Surface Evolver software package[28,73].

## Data availability

The original data underlying Fig. 3a, b, Fig. 4a, c, Fig. 5a–c, and Supplementary Figs. 2-2b, c are provided as Supplementary Movies 1–6. All other data are available from the corresponding author upon reasonable request.

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

## Acknowledgements

We thank Prof. Youhei Kawabata (Rakuno Gakuen University) for the Raman spectrum measurements. This work was supported by JSPS KAKENHI (Grant Number JP20H00120 and JP22K20346) and JSPS Fellow (Grant number 21J11287 to M.K.). M.K. was supported by International Joint Graduate Program in Materials Science at Tohoku University.

## Author contributions

M. K., M. I., and P. W. conceived the work, designed the experiments, and wrote the manuscript with input from all other authors. M. K. conducted experiments and analysis. Y. S. developed the vesicle division induced by adding second amphiphile. R.K. and T. K. supported theoretical part of this study.

## Competing interests

The authors declare no competing interests.
