## [Peer Review File · Communications Chemistry]

Reviewers' comments:

Reviewer #1 (Remarks to the Author):

The paper by Kurisu et Al. is a well-written contribution in the field of synthetic biology and certainly deserves to be published in "Communications Chemistry". The authors devised a synthetic process which is able to mimic the recursive growth & division process, typical of the biological cells. All the aspects related with the chemistry and the physics of the system were investigated in details, as extensively reported in the SI. The manuscript and the figures therein are concise and informative and the text is logically organized.

Therefore, my suggestion is to publish the paper after a few aspects will be clarified in a revised version of the manuscript:

1) In lines 360 – 380 the authors reported some thermodynamic considerations to analyze the formation of para NC PANI-ES and its interactions with AOT molecules. However, the entire paragraph is quite out of context and it is not logically related with the rest of the discussion. I would suggest to better contextualize this section and add a few lines to explain why it is important.

2) The spontaneous curvature model predicts that an equilibrium limiting pear shape is obtained when the reduced volume decreases and the spontaneous curvature (or the leaflets area difference) increases, as also reported by the authors at line 418. However, while the uptake of AOT molecules explains the effect on the curvature of the membrane, it is not clear (at least to me) how the volume loss is attained. Rather, it seems that the osmolarity difference between the lumen of the vesicles and the outer solution always promotes a volume increase, as also reported by the authors (paragraph starting at the line 446). Please, could you clarify this aspect?

3) After the equilibrium pear limiting shape is reached, an extra energy input is generally required to break the neck and let the daughter vesicles divide. The authors explain the spontaneous division in terms of amphiphiles segregation (lines 429 - 445); however, it is not clear to me how, once DLPE and AOT are segregated in the neck region (and hence have reached an equilibrium), this can provide the extra energy necessary for the division step. Is the continuous mass transfer of AOT from the solution to the membrane that keep the system far from equilibrium? Or the enzymatic reaction (and polymerization reaction) can play a role also in this aspect?

Reviewer #2 (Remarks to the Author):

The manuscript by Kurisu and colleagues describes an improved synthetic cell vesicle biomanufacturing system with an energy production unit.

The manuscript is very well written, the rationale is clear and all experiments are described very well. Data is presented clearly and comprehensively. The figures and descriptions are easy to follow and interpret. It appears that all experiments have good controls and appropriate amount of replicates.

The energy production using glucose oxidase was combined with a previously reported vesicle production system consisting of PANI-ES (information polymer) and AOT, resulting in the successful demonstration of recursive vesicle reproduction with an artificial metabolic pathway. I think this work will be of great interest to the biomanufacturing and the synthetic cell engineering community.

In the method, the authors used GOD solution to trigger the polymerization of aniline with AOT LUCs; however, they changed to D-glucose solution for the later aniline polymerization experiment. Is there any specific reason for the change? It would help to interpret and evaluate the results if this was clarified.

Is there any dependence on liposome composition? Given the requirement for membrane permeability, I would expect that this system would depend on type and ratios of lipids used to make the liposomes. It would be very helpful if authors discuss that.

Would this system work with most common vesicle composition used in synthetic cell engineering, POPC/cholesterol? It would be valuable to add an experiment demonstrating this, or discuss why it would not work.

Reviewer #1 (Remarks to the Author):

The paper by Kurisu et Al. is a well-written contribution in the field of synthetic biology and certainly deserves to be published in “Communications Chemistry”. The authors devised a synthetic process which is able to mimic the recursive growth & division process, typical of the biological cells. All the aspects related with the chemistry and the physics of the system were investigated in details, as extensively reported in the SI. The manuscript and the figures therein are concise and informative and the text is logically organized.

We deeply appreciate your understanding of our study and valuable comments. We have now revised our manuscript paying particular attention to your points and the changes in the text were marked using red font. In the following we summarized our responses to your comments.

Therefore, my suggestion is to publish the paper after a few aspects will be clarified in a revised version of the manuscript:

1) In lines 360 – 380 the authors reported some thermodynamic considerations to analyze the formation of para NC PANI-ES and its interactions with AOT molecules. However, the entire paragraph is quite out of context and it is not logically related with the rest of the discussion. I would suggest to better contextualize this section and add a few lines to explain why it is important.

The heart of our synthetic minimal cell is the coupling between the synthesis of an information polymer (PANI-ES) and vesicle reproduction, which originates in specific interactions between the aniline radical cation and the sulfonate head group of AOT. This resembles the Watson-Crick base pairing in the proliferation scheme of living cells. Therefore, we consider that it is important to identify the thermodynamic conditions for the stable encoding of the specific sequences of the polymer. We added the above explanation in the main text (ll. 362-366).

2) The spontaneous curvature model predicts that an equilibrium limiting pear shape is obtained when the reduced volume decreases and the spontaneous curvature (or the leaflets area difference) increases, as also reported by the authors at line 418. However, while the uptake of AOT molecules explains the effect on the curvature of the membrane, it is not clear (at least to me) how the volume loss is attained. Rather, it seems that the osmolarity difference between the lumen of the vesicles and the outer solution always promotes a volume increase, as also reported by the authors (paragraph starting at the line 446). Please,

could you clarify this aspect?

First, we would like to clarify the definition of the reduced volume, $v = V/[(\frac{4\pi}{3})R_0^3]$ ($R_0 = \sqrt{A/4\pi}$; V being the vesicle volume and A being the vesicle surface area) (see ll. 417 – 420 in the main text). The change in the reduced volume occurs as a result of a competition between the increase in the membrane area and that in the inner volume. Therefore, the reduced volume can be decreasing even if the inner volume is increasing. According to the spontaneous curvature model, the vesicle shape is determined by the reduced volume, v , and the reduced spontaneous curvature $c_0 = C_0R_0$ (C_0 , spontaneous curvature). To attain the limiting pear shape vesicle, the values of v and c_0 of the vesicle need to lie on the L^{pear} line in the phase diagram predicted by the spontaneous curvature model (Fig. R1). For example, a symmetric limiting shape vesicle has $v = 0.7$ and $c_0 = 3$ (point L in Fig. R1). In our synthetic minimal cell, the incorporated AOT molecules first stay in the outer leaflet and then move to the inner leaflet by flip-flop motions. The balance between the uptake rate and the flip-flop rate determines the spontaneous curvature and membrane area growth rate. If in the absence of an osmotic pressure difference between the inside and outside of the vesicle, *i.e.*, in the case the volume does not change, the reduced volume decreases as the membrane grows. When the volume changes due to an osmotic pressure difference, the reduced volume is determined by the balance between the area growth rate and the volume growth rate. Thus, to attain the vesicle deformation to proceed to the limiting shape, we regulated the area growth rate by tuning the polymerisation rate for PANI-ES and the supply rate of AOT micelles and mixed SDBS + Chol micelles by adjusting the micro-injection pressure manually (see ll. 478 – 490 in the main text).

Figure R1. Phase diagram based on the spontaneous curvature model.

3) After the equilibrium pear limiting shape is reached, an extra energy input is generally required to break the neck and let the daughter vesicles divide. The authors explain the spontaneous division in terms of amphiphiles segregation (lines 429 - 445); however, it is not clear to me how, once DLPE and AOT are segregated in the neck region (and hence have reached an equilibrium), this can provide the extra energy necessary for the division step. Is the continuous mass transfer of AOT from the solution to the membrane

that keep the system far from equilibrium? Or the enzymatic reaction (and polymerization reaction) can play a role also in this aspect?

To attain spontaneous vesicle division, it is necessary that 1) the two-vesicle state after division is more stable than the limiting shape state before division, and 2) the energy barrier between the two states – limiting shape and two separate vesicles – can be overcome with the thermal energy. The free energy difference between the limiting shape state, F_1 , and the two-vesicle state (two vesicles having the same radius R), F_2 , is expressed by

$$F_2 - F_1 \cong -4\pi a_{ne} \kappa \left[C_0 - \frac{2}{R} \right] + 4\pi \kappa_G,$$

where a_{ne} is the neck radius, κ is the bending rigidity, and κ_G is the Gaussian bending rigidity [M. Imai, et al., *Soft Matter* 18, 4823 (2022)]. Since the Gaussian bending rigidity is estimated as $\kappa_G \sim -\kappa < 0$ and $c_0 = C_0 R = 3$, we obtain $F_2 - F_1 \sim -4\pi \kappa < 0$. Thus, the two-vesicle state is more stable than the limiting shape state. In this paper we show that the segregation of AOT and DLPE (an inverse cone shaped lipid) occurs in the neck region due to the coupling between Gaussian curvature and local lipid composition. This segregation produces an interface between the neck region and the spherical cap region. This interface might reduce the energy barrier and destabilises the neck, causing the vesicle division without extra energy input. Unfortunately, we cannot show this reduction of the energy barrier analytically, but our coarse-grained molecular dynamics simulation study clearly shows that the inverse cone shaped lipid induces the spontaneous vesicle division [N. Urakami, et al., *Soft Matter*, 14, 3018 (2018)]. We added the above explanation in the main text (ll. 436-462).

We have done our best to rewrite our article according to your suggestions. In addition, we tried to optimize the manuscript at several places without any alterations of the data already presented in the original version of the manuscript. We hope that with the changes made the achievements obtained in the work are better described so that a broad readership more clearly understands the concepts of the work and the key findings. We appreciate your kind and critical comments very much and hope our paper will be acceptable as revised.

Best regards and kindest wishes,

Masayuki Imai

Department of Physics, Tohoku University

Reviewer #2 (Remarks to the Author):

The manuscript by Kurisu and colleagues describes an improved synthetic cell vesicle biomanufacturing system with an energy production unit.

The manuscript is very well written, the rationale is clear and all experiments are described very well. Data is presented clearly and comprehensively. The figures and descriptions are easy to follow and interpret. It appears that all experiments have good controls and appropriate amount of replicates.

The energy production using glucose oxidase was combined with a previously reported vesicle production system consisting of PANI-ES (information polymer) and AOT, resulting in the successful demonstration of recursive vesicle reproduction with an artificial metabolic pathway.

I think this work will be of great interest to the biomanufacturing and the synthetic cell engineering community.

We deeply appreciate your understanding of our study and valuable comments. We have now revised our manuscript paying particular attention to your points and the changes in the text were marked using red font. In the following we summarized our responses to your comments.

In the method, the authors used GOD solution to trigger the polymerization of aniline with AOT LUCs; however, they changed to D-glucose solution for the later aniline polymerization experiment.

Is there any specific reasons for the change? it would help to interpret and evaluate the results if this was clarified.

The GOD solution was used to trigger the polymerisation of aniline in the presence of AOT LUVs for the reaction run in a reaction tube, whereas the D-glucose solution was used in the experiments with AOT GUVs by the double micro-injection technique. In both reactions, H₂O₂ was produced once the missing component for the reaction to occur was added. There is no difference between the two experiments in terms of reaction kinetics of H₂O₂ production. Important was to prevent an undesirable osmotic shock to the vesicles. In the micro-injection experiment to achieve membrane growth of the GUVs coupled with the polymerisation of aniline (see Fig.4a), the 20 mM NaH₂PO₄ solution with which the GUVs were prepared contained 100 mM D-sucrose in both the inside and outside of the vesicle membrane. Therefore, when a tiny amount of D-glucose solution (100 mM, ~0.10 nL/sec) was supplied to GUVs to trigger the polymerisation reaction, there was almost no difference in osmolarity across the membrane (Fig. R2a). In fact, volume changes of GUVs were not observed with this experimental setup (Fig. 4c). However, when a D-glucose solution was added to a glucose-free mixture to trigger the polymerisation of aniline in the presence of AOT LUVs (Fig. 2a), the addition of a large amount of D-glucose solution (>100 mM, >100 μL to 0.5 mL) would have caused a non-negligible osmotic pressure difference across the LUV membranes (~100 mOsm/L). Thus, we prepared AOT LUVs in a solution containing 100 mM D-glucose and triggered the reaction by adding the GOD solution instead (Fig. R2b). This is just a replacement of the order of adding the reaction components without affecting the outcome of the reaction. We added the above explanation in the main text (ll. 629-637).

Figure R2. Osmolarity in GUV (a) and LUV (b) systems.

Is there any dependence on liposome composition? Given the requirement for membrane permeability, I would expect that this system would depend on type and ratios of lipids used to make the liposomes. It would be very helpful if authors discuss that.

Would this system work with most common vesicle composition used in synthetic cell engineering, POPC/cholesterol? It would be valuable to add an experiment demonstrating this, or discuss why would it not work.

In our minimal cell, the membrane molecules must satisfy the following three requirements: 1) vesicle membrane growth coupled with an information molecule, 2) deformation of the initial vesicle to the limiting

shape vesicle and division of the limiting shape vesicle, and 3) inflation of the volume of the obtained vesicles to the volume of the initial vesicle. Condition 1 requires that the vesicle-forming amphiphile must have a sulfonate head group (R-SO_3^-). A high critical vesiculation concentration (cvc) is also required for amphiphiles to be incorporated into the vesicle membrane. Condition 2 requires the introduction

of inverse cone-shaped membrane molecules as the second component of the membrane. In addition, the balance between the uptake rate and the flip-flop rate of the membrane molecules determines the deformation to the limiting shape vesicle. The uptake rate is governed by the amount of synthesized PANI-ES segments, while the flip-flop rate is governed by the hydrophobicity of the membrane molecule (balance between hydrophilic and hydrophobic parts). Condition 3 requires regulating the permeabilities of water and osmolytes by adjusting the structure of the hydrophobic part of the membrane molecules. Therefore, as reviewer #2 pointed out, the properties of this minimal cell depend on the type and composition of membrane molecules.

Condition 1 has already been discussed in the previous paper [M. Kurisu, et al., *Commun. Chem.* 2, 117 (2019)]. Please see Table. R1 (Table 2 in M. Kurisu, et al., *Commun. Chem.* 2, 117 (2019)). It should be noted that the cvc of the AOT suspension used in this study is 1.5 mM, which is much larger than the cvc of phospholipids, *e.g.*, the cvc of DPPC is about 1 nM or less. When we supply high-cvc amphiphiles to a target GUV, much of them are delivered to the vicinity of the GUV in their non-associated state or as small aggregates. On the other hand, when we supply low-cvc amphiphiles such as DPPC, most of them are delivered in their self-assembled vesicle state. Therefore, in this latter case other ways to incorporate the amphiphiles into the GUVs must be considered, such as vesicle-vesicle fusion. In this paper we show that the AOT(+SDBS) +Chol or DLPE system satisfies condition 2. Regarding condition 3, we have only results for the AOT (+SDBS) + Chol system. We consider further developing our minimal cell by using other membrane molecules and compositions as next step of the work. Such work is beyond the scope of this paper. We added the above explanation in the main text (*ll.* 491-501).

It would certainly be great if we could expand our minimal cell to a POPC/Chol system, but unfortunately the membrane molecules in our minimal cells must have sulfonate head groups as shown in Table R1, and POPC/Chol system cannot satisfy the above conditions.

Table R1. Observed relative growth of four types of GUVs

GUVs with aniline ^a and HRP ^b	Microinjected suspension/solution, containing H ₂ O ₂ ^c			
	SUVs			micelles
	DOPC	DOPA	AOT	SDBS
DOPC GUV	0	0	0	0
DOPA GUV	0	0	0	0
AOT GUV	0	0	1.0	1.1
SDBS/DA GUV	0	0	0.2	0.9

GUVs were prepared from DOPC, DOPA, AOT or SDBS/DA (1:1), all at 3.0 mM total amphiphile concentration in 20 mM NaH₂PO₄ solution, pH = 4.3, upon micro-injection of 20 mM DOPC SUVs, DOPA SUVs, AOT SUVs, or 100 mM SDBS micelles. The observed growth rate of the GUV is normalised by that of the AOT GUV/AOT SUV system [$d(A(t)/A(0))/dt = 0.0086 \text{ s}^{-1}$]. The growth rates were estimated from the growth stages in Fig. 3b; i.e., 10-40 s for AOT to AOT, 60-90 s for SDBS to AOT, 10-100 s for AOT to SDBS/DA, and 10-50 s for SDBS to SDBS/DA
^a[Aniline] = 4.0 mM
^b[HRP] = 0.92 μM
^c[H₂O₂] = 2.0 M

We have done our best to rewrite our article according to your suggestions. In addition, we tried to optimize the manuscript at several places without any alterations of the data already presented in the original version of the manuscript. We hope that with the changes made the achievements obtained in the work are better described so that a broad readership more clearly understands the concepts of the work and the key findings. We appreciate your kind and critical comments very much and hope our paper will be acceptable as revised.

Best regards and kindest wishes,

Masayuki Imai

Department of Physics, Tohoku University

REVIEWERS' COMMENTS:

Reviewer #1 (Remarks to the Author):

I found the Authors' answers to both reviewers' comments convincing. I think the revised manuscript can be accepted for publication.

Reviewer #2 (Remarks to the Author):

The authors sufficiently addressed all questions and comments raised in the review. The changes in the paper are very good, and I have no other questions or revision requests.